# Dynamic beam steering with all-dielectric electro-optic III–V multiple-quantum-well metasurfaces

Pin Chieh Wu [1,2], Ragip A. Pala[1], Ghazaleh Kafaie Shirmanesh[1], Wen-Hui Cheng [1], Ruzan Sokhoyan [1], Meir Grajower[1], Muhammad Z. Alam[1], Duhyun Lee[1,3] & Harry A. Atwater [1,4]

Tunable metasurfaces enable dynamical control of the key constitutive properties of light at a subwavelength scale. To date, electrically tunable metasurfaces at near-infrared wavelengths have been realized using free carrier modulation, and switching of thermo-optical, liquid crystal and phase change media. However, the highest performance and lowest loss discrete optoelectronic modulators exploit the electro-optic effect in multiple-quantum-well hetero-structures. Here, we report an all-dielectric active metasurface based on electro-optically tunable III–V multiple-quantum-wells patterned into subwavelength elements that each supports a hybrid Mie-guided mode resonance. The quantum-confined Stark effect actively modulates this volumetric hybrid resonance, and we observe a relative reflectance modulation of 270% and a phase shift from 0° to ~70°. Additionally, we demonstrate beam steering by applying an electrical bias to each element to actively change the metasurface period, an approach that can also realize tunable metalenses, active polarizers, and flat spatial light modulators.

[1] Thomas J. Watson Laboratory of Applied Physics, California Institute of Technology, Pasadena, CA 91125, USA. [2] Department of Photonics, National Cheng Kung University, Tainan 70101, Taiwan. [3] Samsung Advanced Institute of Technology, Suwon, Gyeonggi-do 443-803, South Korea. [4] Kavli Nanoscience Institute, California Institute of Technology, Pasadena, CA 91125, USA. Correspondence and requests for materials should be addressed to P.C.W. (email: pcwu@gs.ncku.edu.tw) or to H.A.A. (email: haa@caltech.edu)

Achieving versatile dynamical control of the key constitutive properties of light at the nanoscale is a grand challenge for nanophotonics. In the last several years, metasurfaces have shown extraordinary promise to achieve such comprehensive control over the characteristics of scattered light. Metasurfaces can be viewed as artificially designed arrays of subwavelength optical scatterers, where each scatterer introduces abrupt changes to the phase, amplitude, or polarization of the reflected or transmitted electromagnetic waves[1–3]. Thus, metasurfaces offer the ability to control the wavefront of the scattered light, thereby creating new flat optics and ultrathin optoelectronic components[4,5]. To date, metasurfaces have been used to demonstrate a number of low-profile optical components with important capabilities, including focusing[6–9], polarization control and detection[10–12], holograms[13–15], and quantum light control[16,17].

Among the large volume of experimental reports about metasurfaces, most demonstrated so far are passive. For passive metasurfaces, the light-scattering characteristics are defined by the geometry and arrangement of subwavelength scatterers, fixed at the time of fabrication. In contrast to passive metasurfaces, actively tunable metasurfaces can realize multiple functions[18–20], serving as low-profile nanophotonic devices capable of beam steering, active polarization switching, and formation of reconfigurable metalenses.

So far, a number of different methods have been used to realize tunable metasurfaces, commonly by incorporating an active material into the metasurface structure. The dielectric permittivity of the active material is then dynamically controlled via application of an external stimulus, such as an electrical bias[21,22], laser pulse[23], or heat input[24]. Reconfigurable metasurfaces, which are based on incorporating active materials into otherwise passive antenna arrays, are hereafter referred to as hybrid metasurfaces. For example, incorporation of monolayer graphene into a plasmonic metasurface can enable active tuning of the spectral response by electrically tuning the Fermi level of the graphene sheet[25–27]. Electrical tuning of the coupling between metasurface resonances and intersubband transitions in multiple-quantum-wells (MQWs) has also been explored for applications, such as tunable filters[28] and optical modulators at mid-infrared wavelengths[29,30]. To achieve active metasurface performance at visible and near-infrared (NIR) wavelengths, the integration of metasurfaces with phase-change materials or liquid crystals has enabled the demonstration of phase modulation[31] and active beam switching[32,33]. Modulation of the dielectric permittivity near the epsilon-near-zero (ENZ) transition in doped transparent conducting materials can yield large optical modulation of the scattered light[34], and to date the ENZ transition in indium tin oxide[35–37] and titanium nitride[38] has been exploited to electrically tune the properties of scattered/emitted light. These hybrid metasurfaces[34–37] operate by spectrally overlapping the geometrical antenna resonance and the ENZ permittivity regime, and also spatially overlapping the metasurface element mode profile with the tunable permittivity transparent conducting material. To enable a widely tunable optical response, strong local field confinement and enhancement in the active material is required. Prior research has also combined tunable metasurface optics with microelectromechanical systems (MEMS) technology to demonstrate varifocal lenses[39]. Moreover, previous work has shown that fabricating metasurfaces on elastomeric substrates may yield adaptive metalenses[40], strain-multiplexed meta-holograms[41], and an active control of the structural color[42]. However, in MEMS-based and mechanically stretchable substrate modulation approaches, control of the optical response is achieved by changing the distance between either adjacent metasurface elements[43,44] or entire element arrays[39], and requires a

mechanical transducer, which limits the frequency bandwidth. While interesting, these approaches are not able to yield versatile active control over the scattered light wavefront over a wide frequency range. This condition can only be realized by electronic tuning the optical response of each metasurface element.

Fabricating metasurface elements directly in an active material could substantially simplify the metasurface design and facilitate the fabrication process. For example, prior research has used phase-change materials as metasurface building blocks to achieve actively tunable optical responses[19,45,46]. The ability to rewrite metasurface patterns incorporating phase-change materials with a pump laser has enabled the demonstration of multiple functions when using a single sheet of either GST or $VO_2$[19,46]. However, the tuning speed of the phase-change-material-based tunable metasurfaces is usually slow, because the phase transition speed is typically limited by the thermal response time in material heating.

Previously, a GaAs all-dielectric tunable metasurface[23] has been realized by actively refractive index tuning resulting from free carrier generation via an optical pump[23]. This approach enables a picosecond response time, but the requirement of an ultrafast pump laser beam is not desirable for many low-power compact nanophotonic applications. Under optical pumping, the area for refractive-index modulation is determined by the size of focused laser spot, and is relatively large, limiting the possibility to achieve control of individual metasurface elements. To achieve an independent control of each metasurface element, it is preferable to modulate the optical response of the metasurface electrically rather than optically. Prior research has shown that a patterned graphene layer under applied bias voltage can be used to actively modulate the properties of the scattered light[47,48]. So far, the actively tunable optical response of the patterned graphene layer has only been demonstrated in the mid-infrared wavelength regime because of the achievable carrier densities in doped graphene. Thus it remains an outstanding research challenge to develop an active metasurface platform operating in the visible or NIR wavelength range that would dynamically tailor the wavefront of scattered light by modulation of individual antenna elements.

Here, we describe an electrically tunable metasurface, which utilizes III–V compound semiconducting MQW structures as resonant elements. The amplitude and phase of the light reflected from the metasurface can be continuously tuned by applying a DC electric field across the MQW metasurface elements, with a tunable optical response from the quantum-confined Stark effect (QCSE)[30,49]. The QCSE enables electro-optic modulation of the MQW complex refractive index, most strongly at wavelengths near the MQW bandgap. In our metasurface design, each MQW resonator supports a hybrid-resonant mode with a relatively high-quality factor, enabling optical modulation under applied bias. Using this active device concept, we experimentally demonstrate beam steering by electrically controlling the optical response of individual metasurface elements. The QCSE is widely used in high-performance electro-optical components such as high-speed modulators[50]. Thus, our approach combines the well-established MQW technology with subwavelength antennas to creating an active metasurface platform for diverse nanophotonics applications.

## Results

**Characterization of MQW.** We utilize an epitaxial III–V heterostructure consisting of a GaAs substrate, distributed Bragg reflector (DBR) and a 1.23-μm-thick undoped MQW layer (see inset of Fig. 1a). The DBR is comprised 20 pairs of alternating layers of n-doped $Al_{0.9}Ga_{0.1}As$ (76.5 nm) and n-doped GaAs (65 nm) with the n-doped $Al_{0.9}Ga_{0.1}As$ as the topmost layer. A

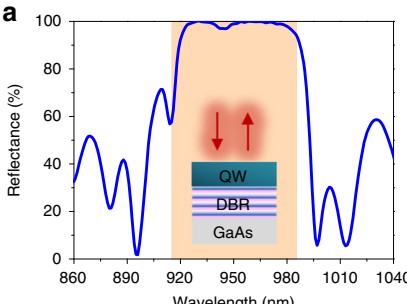
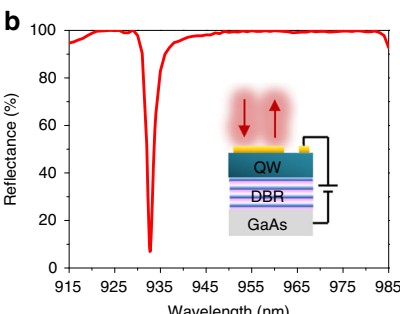
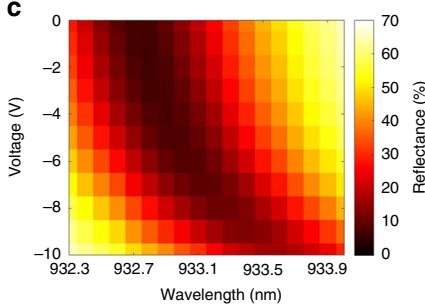

**Fig. 1** Optical performance of MQWs. Measured reflectance spectra of (**a**) a bare DBR/MQW and (**b**) a DBR/MQW/Ti/Au Fabry–Pérot cavity under no applied bias. The insets show the schematics of corresponding structures. The light orange area in (**a**) indicates the wavelength range shown in (**b**). **c** Measured reflectance of the DBR/MQW/Ti/Au Fabry–Pérot cavity as a function of wavelength and applied voltage

50- nm-thick p-doped GaAs contact layer with a carrier density of $10^{19}$ cm$^{-3}$ is grown on top of the MQWs (not shown in the inset of Fig. 1a). Figure 1a illustrates the measured reflectance spectrum of the planar MQW/DBR/GaAs structure. As seen in Fig. 1a, for wavelengths ranging from 915 nm to 990 nm, the reflectance is close to 100%, indicating that the DBR acts as a high-quality mirror in this wavelength range. We also observe a sharp reflectance dip at a wavelength of ~915 nm. This reflectance dip originates from near-bandgap absorption in the MQW layer. As a next step, we investigate the tunable optical response of the MQW layer. Due to the QCSE, the interband transition energy is shifted by applying a DC electric field across the quantum wells resulting in bias-induced MQW complex refractive index modulation[49]. For our quantum well heterostructures (Supplementary Fig. 1a), the expected modulation of the real part of the refractive index is on the order of $\Delta n = 0.01$[51]. To be able to experimentally observe this small variation of the real part of the refractive index, we integrated a Fabry–Pérot resonant cavity around the MQWs whose top mirror was formed by depositing a 35-nm-thick semitransparent Au film on top of the MQWs. To improve the adhesion of Au to the top p-doped GaAs layer, we first deposited a 2-nm-thick Ti film before depositing the Au film (see the inset of Fig. 1b). Figure 1b plots the measured reflectance spectrum of the fabricated DBR/MQW/Au Fabry–Pérot cavity. As seen in Fig. 1b, the Fabry–Pérot cavity exhibits a narrow resonance at a wavelength of 932.7 nm. This narrow resonance enhances the optical modulation caused by the variation of the complex refractive index of the MQWs under applied bias.

Once we measured the reflectance spectrum of the DBR/MQW/ Au planar heterostructure, we then measured its reflectance modulation under applied bias. To facilitate bias application, we deposited ohmic contacts on the topmost p-doped GaAs layer [Ti (20 nm)/Pt (30 nm)/Au (300 nm)] and at the bottom of the n-doped GaAs substrate [Ge (43 nm)/Ni (30 nm)/Au (87 nm)]. We then measured the reflectance from our Fabry–Pérot resonant MQW sample, when the GaAs substrate (low potential) and the top ohmic contact (high potential) are biased with respect to each other (see the inset of Fig. 1b). Figure 1c shows the map of the measured reflectance as a function of wavelength and applied bias. When the external bias is applied, we observe a shift of the resonant wavelength that is accompanied with a significant reflectance modulation. This demonstrates that both the real and imaginary part of the refractive index of the MQWs are modulated by applied bias. To study the tunable optical response of the MQWs at different wavelengths, the position of the Fabry–Pérot resonance has to be shifted to the desired spectral position. We achieved this by spin coating a thin PMMA layer with a pre-defined thickness between the Au film and the MQW layer so as vary the cavity length. The spectral position of the Fabry–Pérot resonance thus varies with the thickness of the

PMMA spacer layer in the cavity (Supplementary Fig. 1b). Our analysis shows that, as expected, larger optical modulation is observed at shorter operation wavelengths near the semiconductor band edge (Supplementary Note 1). From these measurements, we concluded that the optimal wavelength to observe a large reflectance modulation was between 915 nm and 920 nm.

**Design and characterization of all-dielectric tunable metasurface.** Once we had identified the optimal operation wavelength for observing the tunable optical response of MQWs, we designed and fabricated our tunable metasurface. Since our MQWs exhibit relatively modest refractive index change under applied bias, the designed metasurface element has to support high-quality resonant mode near the semiconductor band edge to exhibit significant optical modulation under applied bias. The fundamental electric or magnetic dipole modes of typically utilized dielectric resonators do not possess sufficiently high-quality factors. Figure 2a shows the schematic of our electrically tunable all-dielectric III–V MQW resonator-based metasurface. The resonator design has a double-slit structure, where the double slits have been partially etched into the MQW layer (inset of Fig. 2b). We choose the structural parameters of our resonators such that these slits support a guided mode (GM) resonance that hybridizes with a higher-order Mie resonance, at a wavelength slightly beyond the MQW band edge absorption wavelength. The geometrical parameters of the metasurface elements are summarized in the caption of Fig. 2. The inherently large real part of the refractive index ($n \approx 3.62$) of our MQWs enables us to design subwavelength resonators (metasurface elements), which are only 700 -nm wide. The simulated reflectance spectrum of our metasurface is shown in Fig. 2b. As seen in Fig. 2b, the metasurface supports two distinct resonant modes at wavelengths of 915.9 nm and 936.3 nm. Figure 2c and d shows the z-component of electric- and magnetic field intensities in our metasurface element at both resonant wavelengths. The calculated field profiles show that at a wavelength of 915.9 nm, the metasurface element supports a high-order Mie resonance (left images in Fig. 2c, d). The multipole decomposition analysis[52–54] shows that the supported high-order Mie resonant mode is dominated by the magnetic octupolar mode (Supplementary Note 2). In addition, at the same wavelength, our metasurface element supports a GM resonance propagating along x direction, resulting in an electric field that leaks into the air gaps separating the metasurface elements (Supplementary Note 3). Hence, the resonant mode supported by the metasurface element at a wavelength of 915.9 nm can be interpreted as a coupling of a Mie resonance and a GM resonance, which is referred to here as a hybrid Mie-GM resonance. Note that the coupling of two resonant modes normally results in mode splitting. In our case, the mode splitting can be seen when

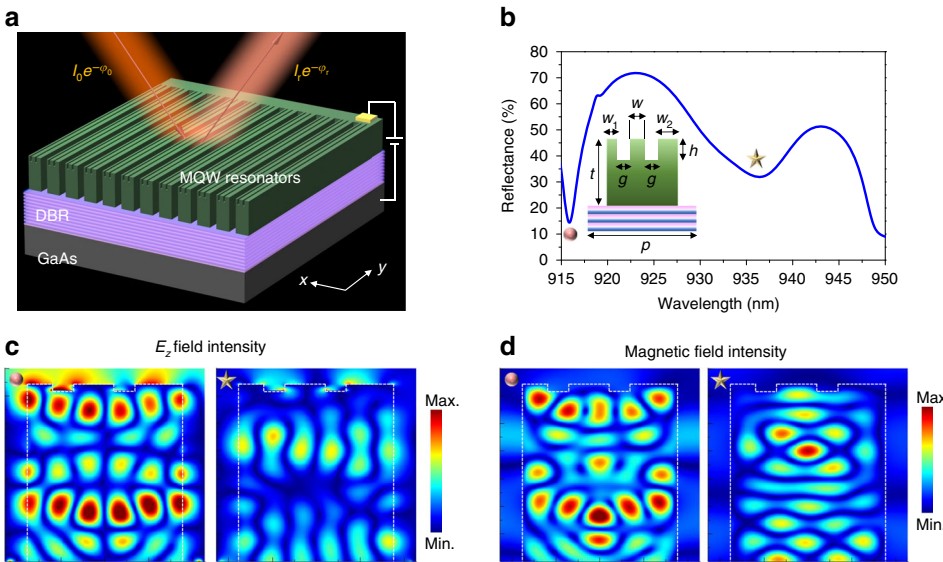

**Fig. 2** Resonant modes in MQW-based metasurface. **a** Schematic for all-dielectric MQW metasurface. The metasurface consists of a n-doped GaAs substrate, a distributed Bragg reflector (DBR), and a 1230-nm-thick MQW layer. There is a 50-nm-thick p-doped GaAs layer with doping level of $10^{19}$ cm$^{-3}$ that is grown on top of the MQWs as a top contact. **b** Simulated reflectance spectrum of a Mie-GM resonant metasurface under an $x$-polarized normal illumination. The inset shows the schematic of the metasurface unit element. The unit element dimensions are defined as follows: $w_1 = 110$ nm, $w_2 = 210$ nm, $w = 180$ nm, $g = 100$ nm, $t = 1230$ nm, and $h = 40$ nm. The periodicity $p$ is 910 nm. **c**, **d** show the spatial distributions of the $E_z$ and magnetic field intensities at the wavelengths corresponding to the resonant dips

extending the simulation range to the shorter wavelengths (Supplementary Note 4). We observe another resonance at a wavelength of 936.3 nm, which can be interpreted as a Fabry–Pérot resonance coupled to a GM resonance (Fig. 2c; Supplementary Fig. 7b). The Fabry–Pérot resonance-like mode propagates along $z$ direction of our 1.23-μm tall resonators and couples with a GM resonance propagating along the $x$ direction of our 700-nm-wide resonators. Full-wave simulations suggest that the GM resonance at the wavelength of 915.9 nm arises mainly from the partially etched double-slit structures, while the one at 936.3 nm is mostly attributable to the MQW slab (Supplementary Note 3).

Next, we fabricated a tunable metasurface and experimentally investigated its tunable optical response. We fabricated our metasurface by electron-beam lithography and inductively coupled plasma-reactive ion etching (ICP-RIE) etching (see the Methods section). Figure 3a shows a scanning electron microscopy (SEM) image, in which double slits are observed on the top of a MQW slab. Figure 3b shows the measured reflectance spectra of the fabricated metasurface under different applied biases (see Supplementary Fig. 9 for details of optical setup). At zero bias, two resonant dips are clearly observed, which is consistent with our simulation results shown in Fig. 2b. When we decrease the applied bias from 0 V to −10 V, we observe a significant red shift of the shorter-wavelength resonance, which corresponds to the hybrid Mie-GM resonance. Moreover, we observe a simultaneous increase of reflectance intensity with decreased bias. Under an applied bias, both the real and imaginary parts of the complex refractive index of the quantum well are modulated. The observed red shift of the resonance indicates an increase of the real part of the refractive index. The modulation of the real part of the refractive index is responsible for the bias-induced phase shift of the reflected light. On the other hand, the change of the reflectance at the resonance dip is caused by the modulation of the imaginary part of the refractive index. Therefore, our hybrid Mie-GM resonance tunable metasurface can be used as an efficient amplitude modulator. When analyzing the behavior of the reflectance at a wavelength of 938 nm, we observe that the

shift of the resonance dip is negligible, while the reflectance at this wavelength is slightly decreased. At wavelengths longer than 940 nm, high-quality resonances are absent and the index change is smaller, so we observe no significant reflectance modulation at these wavelengths. Since our metasurface exhibits much stronger optical modulation at wavelengths corresponding to the hybrid Mie-GM resonance, we focused our characterization on the tunable resonance at shorter wavelengths. To gain further insight, we plot the relative reflectance as a function of wavelength and applied bias (Fig. 3c). The relative reflectance is defined as $[R(V_a \neq 0\,\text{V}) - R(V_a = 0\,\text{V})]/R(V_a = 0\,\text{V}) = \Delta R/R_0$, where $V_a$ is the applied voltage. As mentioned above, in Fig. 3c, we limit the wavelength range between 915 nm and 925 nm. When $V_a$ decreases from 0 V to −10 V, we observe a strong relative reflectance modulation. In particular, at a wavelength of 917 nm, we observe a relative reflectance modulation as high as 270%. The relative reflectance modulation decreases to about −45% at a wavelength of 925 nm. Thus, the proposed III–V MQW resonator-based metasurface is a promising candidate for tunable amplitude modulation. It is worth mentioning that we observe about +20% and −30% absolute reflectance modulation [defined as $R(V_a \neq 0\,\text{V}) - R(V_a = 0\,\text{V})$] at wavelengths of ~917 nm and ~924 nm, respectively (Supplementary Fig. 10). Although they are quantitatively comparable, the phase modulation and desired optical functionality can only be observed when a high-quality factor resonance is present (which shows higher relative reflectance modulation), as can be seen in the following sections. In addition, since the amplitude modulation is achieved via the electro-optic effect rather than charge-carrier injection (due to the low leakage current density in our samples, see Supplementary Note 11), the intrinsic modulation frequency of our device can be MHz (Supplementary Fig. 12) or substantially higher[30,49,50].

We also experimentally evaluated the phase shift of the reflected beam under applied bias at wavelengths of 917 nm and 924 nm using a Michelson interferometer system[31,35,37]. The incident laser spot was positioned to illuminate the edge of the resonator-based metasurface. As a result, part of the incident beam is reflected from the metasurface, while the other part is

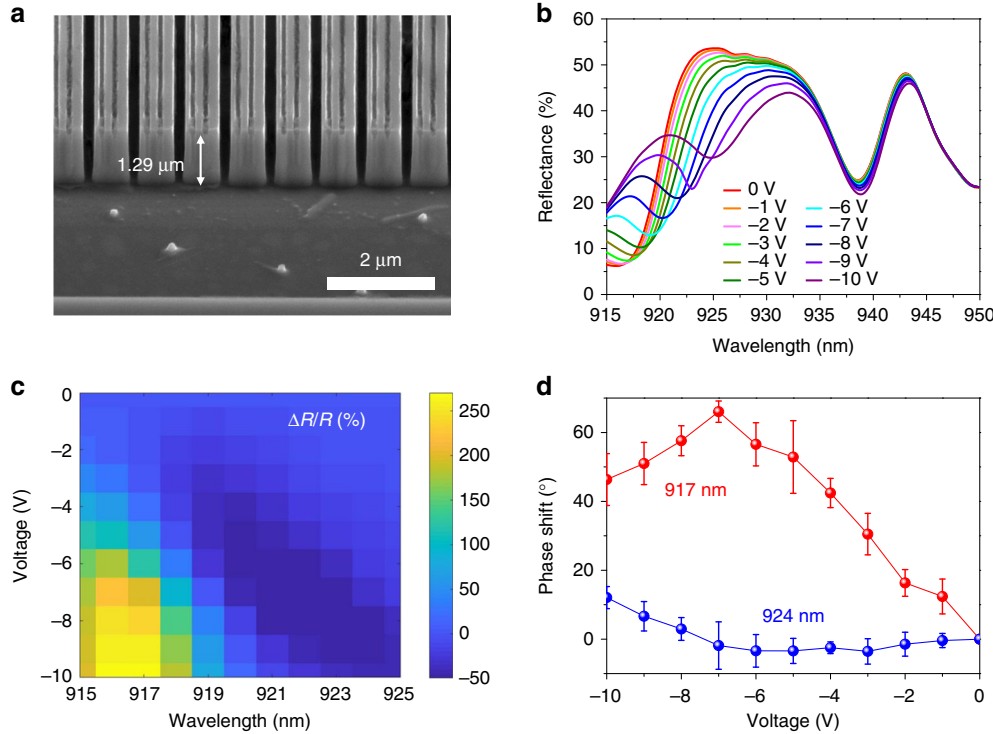

**Fig. 3** Experimental verification of optical modulation in MQW resonators. **a** SEM image of MQW-based hybrid Mie-GM resonators with double slits. **b** Measured reflectance spectra of the resonator array for different applied bias voltages. The incoming light is polarized perpendicularly to the MQW stripes. **c** Measured relative reflectance of the hybrid Mie-GM resonant metasurface as a function of wavelength and applied voltage. We consider the wavelength range from 915 nm to 925 nm with a step of 1 nm. **d** Measured phase modulation at two different wavelengths. Red: 917 nm, blue: 924 nm. Each data point corresponds to an average phase shift measured at four different positions on the sample, while each error bar indicates the standard deviation of the obtained four data points

reflected from the unpatterned MQW heterostructure, and the phase shift was determined using the unpatterned MQW heterostructure as a built-in phase self-reference. By processing and fitting the interference fringes captured by a camera, we are able to calculate the relative displacement of interference fringes between the hybrid Mie-GM resonator region and the unpatterned region. From this, we retrieved the phase shift acquired due to the applied bias. Figure 3d shows the measured phase shift as a function of applied bias at wavelengths of 917 nm (red dots) and 924 nm (blue dots). When the applied bias is decreased from 0 V to −7 V, we observe a continuous increase in phase shift by about 70° at a wavelength of 917 nm. The phase shift decreases to about 50° when the applied bias is further decreased to −10 V. The phase modulation becomes weaker for wavelengths away from the hybrid Mie-GM resonance. For example, at a wavelength of 924 nm, the largest phase shift reduces to 12°, which we observed at an applied bias of −10 V. This modest phase shift is also accompanied by a weaker relative reflectance modulation of −45% (Fig. 3c). These results are consistent with the Kramers–Kronig relation: a large change of the real part of the refractive index (phase shift) is accompanied with a significant modulation of the imaginary part of the refractive index (reflectance modulation).

**Electrical switching of beam diffraction**. As experimentally demonstrated, an extremely strong relative reflectance modulation (~270%) with a phase shift of ~70° can be achieved by electrically biasing the metasurface. As a next step, we patterned the edges of the metasurface to selectively apply a bias to independent groups of metasurface elements, enabling active control of reflectance of the independent element groups. This enables us

to demonstrate an electrically switchable grating resulting in dynamic beam diffraction, which was detected as a far-field radiation pattern. To create the switchable diffraction grating, we fabricated a metasurface with similar structural dimensions as the one described in the inset of Fig. 2b, but we electrically connected in parallel the resonant stripes in groups of three, and leave the adjoining group of three resonant stripes isolated, as shown in Fig. 4a. Under zero applied bias, we observe a single output beam in the Fourier plane that is reflected normally corresponding to the zeroth-order diffracted beam. Higher-order diffracted beams are absent since the period of our metasurface, $p = \Lambda = 910$ nm, is subwavelength at 0 V bias (see details in Supplementary Note 6). When we apply a negative bias voltage, the reflectance of the electrically connected MQW resonators increases, causing an effective increase in the period of the metasurface array ($6 \times p = \Lambda' = 5460$ nm). This increased period creates first-order diffracted beams which appear at an angle defined by the grating equation: $\theta = \sin^{-1}\left(\frac{m\lambda}{p_g}\right)$, where $p_g$ is the period of the grating and $m$ is the diffraction order. Figure 4b shows the SEM image of the fabricated device. Figure 4c shows the schematic optical setup used for measurement of the far-field radiation pattern in the Fourier space. We utilized an uncollimated white light source from a halogen lamp to visualize the sample surface. When measuring the far-field radiation pattern, we use a coherent NIR laser beam (Toptica Photonics CTL 950) as a light source. The laser beam was focused using a long working distance objective with ×10 magnification and 0.28 numerical aperture. The radiation pattern is captured directly by the CCD camera, which is positioned in the Fourier plane. Figure 4d and e shows experimentally measured diffraction patterns for different applied bias voltages. The dynamic diffraction pattern measurements have

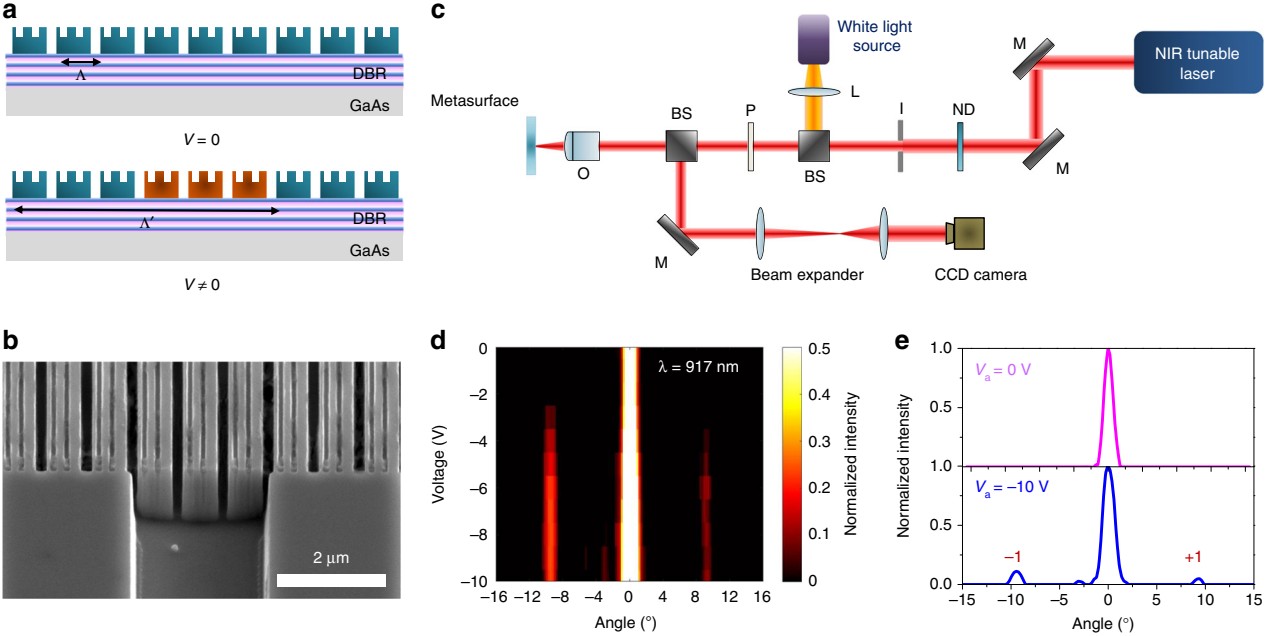

**Fig. 4** Demonstration of switchable diffraction grating. **a** Schematic of the dynamic diffraction grating, which can be realized by changing the grating periodicity via an appropriate bias application. **b** SEM image of the fabricated sample, which we used for the demonstration of the dynamic beam switching. **c** Optical setup for far-field radiation pattern measurements. M: mirror, ND: neutral density filter, I: iris, BS: beam splitter, O: objective with 0.28 numerical aperture, L: lens. **d** Experimentally measured intensity of the scattered light in the far-field as a function of diffraction angle and applied voltage. The first-order diffracted beam appears when the applied voltage is lower than −3 V. The plotted diffracted light intensity is normalized to the light intensity at 0°. The light intensity is plotted for a wavelength of 917 nm. **e** Intensity of the scattered light as a function of the diffraction angle at an applied bias of 0 V (top panel) and at an applied bias of −10 V (bottom panel). The ±1 diffraction order clearly appears when the applied voltage is decreased to −10 V

been performed at a wavelength of 917 nm, which corresponds to the hybrid Mie-GM resonant mode supported by the unit elements of the metasurface. As seen in Fig. 4d, we observe the first-order diffracted beam only in the cases when the contrast of the reflectance between the resonator groups (i.e., the difference in their refractive indices) becomes significantly large. For applied bias voltages between 0 V and −3 V ($0\,V \geq V_a \geq -3\,V$) we observe specular reflection from the metasurface. For applied bias voltages below −3 V ($V_a \leq -3\,V$), the first-order diffracted beam appears at an angle of about 9.66° in the Fourier plane. Interestingly, the intensity of the first-order diffracted beam saturates when the absolute value of the applied bias is lower than −6.5 V (Fig. 4d). We do not observe the first-order diffracted beam when the incident wavelength is switched to 924 nm (Supplementary Note 8). This is expected because there is no significant reflectance and phase modulations at this wavelength.

**Dynamic beam steering with all-dielectric electro-optic MQW metasurface.** Apart from switchable beam diffraction, we experimentally demonstrated beam steering, which requires control over individual metasurface elements. The spatial position of the first diffraction order can be effectively shifted when the periodicity of metasurface is changed, enabling manipulation of far-field radiation. To realize dynamic beam steering, we designed and fabricated another metasurface in which each unit element is electrically isolated by fully etching the air gap between the quantum well slabs. Due to the high refractive index and large thickness of the MQW slabs, the resonant mode is sensitive to minor variations of metasurface structural parameters. This results in different electromagnetic field profiles between the first and the second hybrid Mie-GM metasurfaces (Fig. 2c, d; Supplementary Fig. 14). Our beam steering metasurface also supports a hybrid Mie-GM resonant mode at a wavelength near the band edge absorption of the MQWs, yielding reflectance and phase

modulation (Supplementary Note 9). Figure 5a–d shows the images of the fabricated sample where 64 unit MQW resonator elements are electrically connected to a printed circuit board (PCB) via wire bonding, and each element is independently controlled (see the Methods section). We first examined the spectral response as well as the active optical modulation of this metasurface by applying an identical electrical bias to all the array elements. This beam steering metasurface sample yielded about 80% relative reflectance modulation with a phase shift of ~42° (Supplementary Fig. 15). Next, by individually addressing each metasurface element, we steered the reflected beam, as seen in Fig. 5e, where the first-order diffraction angle becomes smaller as the metasurface periodicity is increased via electrical control. Our numerical simulations show similar far-field radiation patterns (Supplementary Fig. 16a). Note that the sidelobes appear around the zeroth-order diffraction beam are from the finite aperture effect, which can be seen in both measured and simulated results. By characterizing the measured and simulated far-field radiation patterns with a larger total number of unit elements (Fig. 5e; Supplementary Fig. 16), the first-order diffraction peaks can be picked out, as indicated by black arrows in Fig. 5e. In addition, as seen from our simulations, the width of both zero- and first-order diffracted beams is narrower when the total number of unit elements is larger (Supplementary Fig. 16b).

## Discussion
In summary, we have demonstrated an all-dielectric active metasurface platform based on an electro-optic effect in III–V compound semiconducting MQWs. Our metasurface consists of an array of two-dimensional hybrid Mie-GM resonators which exhibit an actively tunable optical response under applied bias in the NIR wavelength range. By applying a DC electric field across the hybrid Mie-GM resonators, we have experimentally observed a relative reflectance modulation of ~270% at a hybrid Mie-GM-

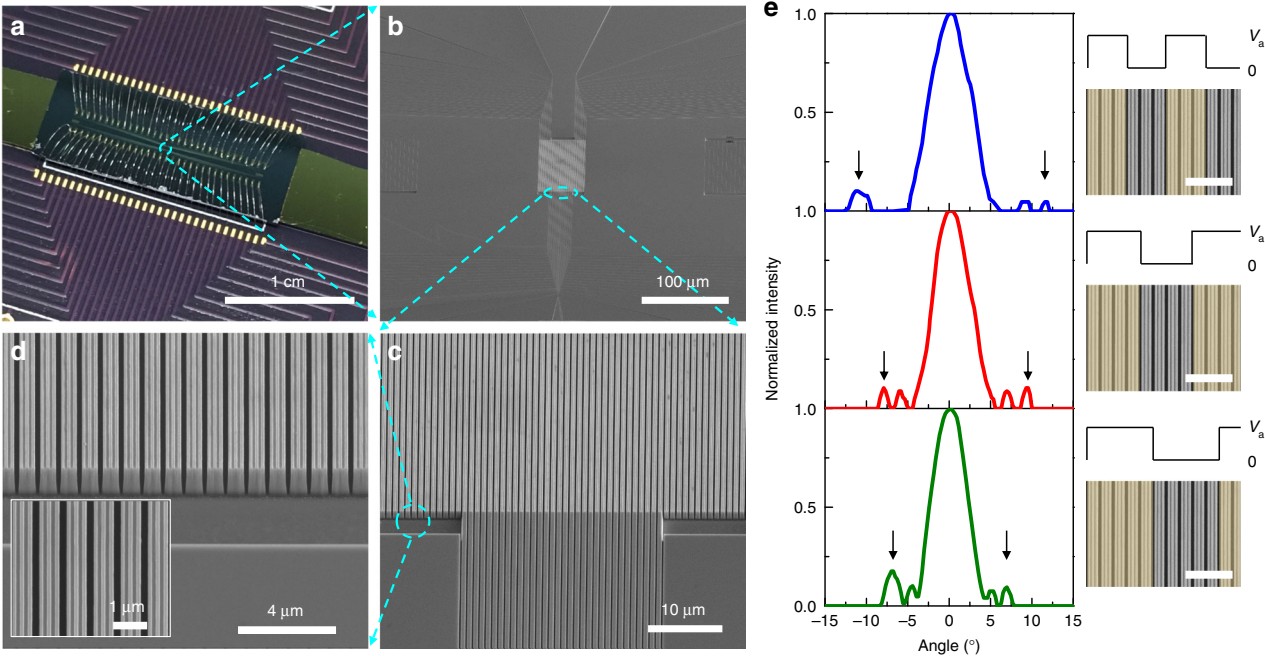

**Fig. 5** Tunable beam steering with MQW-based all-dielectric metasurface. **a** Photographic image of the fabricated gate-tunable metasurface for the demonstration of dynamic beam steering. The metasurface sample is mounted on a voltage-deriving PCB, which has 64 contact pads for individually applying bias on each metasurface unit element. **b**–**d** Scanning electron microscopy images of the gate-tunable metasurface. To electrically isolate every unit structure, portions of the sample are fully etched. Inset in (**d**) shows the top view of the fabricated metasurface. **e** Measured results of dynamic beam steering by electrically changing the periodicity of metasurface. Black arrows indicate the position of the first diffraction order in each case. Right column illustrates how the spatial arrangement of electrical bias changes the periodicity of metasurface. The incident wavelength is fixed at 917 nm. Scale bars: 3 μm

resonant wavelength of 917 nm. We have also measured a continuous phase shift from 0° to 70° at a wavelength of 917 nm. To demonstrate a dynamic diffraction grating utilizing this large reflectance modulation with modest phase shift, we have electrically connected metasurface elements in groups of three and actively changed the metasurface period by applying DC electric field across the hybrid Mie-GM resonators. As a result, we have been able to electrically switch on and off the first-order diffracted beam. Finally, as a proof-of-concept, we further experimentally demonstrate a dynamic beam steering with electrical and individual control of each unit element over the MQW metasurface array.

In our work, our starting point is a monolithically grown III–V compound semiconducting wafer, which we pattern by using electron-beam lithography and dry etching. This is quite different from the case of hybrid active metasurfaces, where the metasurface cannot be grown monolithically. Monolithically grown MQW active metasurfaces can potentially be integrated with existing light emitting devices, such as vertical-cavity surface-emitting lasers (VCSELs). Such an integrated device could serve as a base for future on-chip light detection and ranging systems. Since the tunable optical response of MQWs is based on an electro-optic effect, the proposed metasurface platform also offers the benefit of high modulation speed. The presented active metasurface platform may be useful for the realization of dynamically tunable ultrathin optical components, such as tunable metalenses with reconfigurable focal lengths and numerical apertures, on-chip beam steering devices, active polarizers, and flat spatial light modulators. The performance of the proposed all-dielectric metasurface with hybrid Mie-GM resonance can be further improved by utilizing alternative quantum well systems which exhibit larger modulation of the real part of the refractive index and lower optical

loss as compared to the quantum well used in the present work[55,56] (Supplementary Note 13).

## Methods

**Numerical simulation.** All numerical simulations were carried out using finite difference time domain (FDTD) method (Lumerical). When designing our MQW resonators, we used the perfectly matched layer (PML) boundary condition in $z$ direction and the periodic boundary conditions in $x$ direction. Hence, the calculations of the reflectance spectrum of the MQW resonators were performed in the array configuration. The MQW resonators were assumed infinite in $y$ direction. In our electromagnetic calculations, we assumed that the incoming light impinged normally on the metasurface. That is, the incoming electromagnetic wave propagated along the $z$ direction. For simplicity, the refractive indices of the n-$Al_{0.9}Ga_{0.1}As$, n-$Al_{0.31}Ga_{0.69}As$, $GaAs_{0.6}P_{0.4}$, and InGaP were set as a constant of 3, $3.39 + 0.004i$, $3.3 + 0.004i$, $3.2 + 0.004i$, respectively. The effective refractive index of MQW can be found in Supplementary Fig. 18.

**Sample fabrication.** First, the bottom Ge/Ni/Au Ohmic contact of thickness 43 nm/30 nm/87 nm was deposited on the n-doped GaAs substrate of the MQW wafer by electron beam evaporation. Next, a 1.5-μm-thick 950 PMMA A9 layer was spin coated at 4000 rpm on the front side of the prepared MQW wafer for 60 s. The MQW sample with the PMMA layer on top was then baked on a hot plate for 3 min at 180 °C. Subsequently, we defined the top Ohmic contacts and alignment markers by using the development, metal deposition and lift-off processes where the patterning was done via an electron beam direct write lithography system [VISTEC electron beam pattern generator (EBPG) 5000 +] at an acceleration voltage of 100 keV with a current of 5 nA. After defining the top Ohmic contacts and alignment markers, ZEP 520A was spin coated at 4000 rpm for 60 s, and the sample was then baked on a hot plate for 3 min at 180 °C. The double-slit structures were defined via the electron beam writing system at an acceleration voltage of 100 keV with a current of 0.3 nA. The sample was then baked for 2 min at 110 °C and developed at about 10 °C for 90 s. The structured ZEP 520 A was used as a mask for dry etch process, which was employed for fabrication of double slits. The etching was performed using a III–V compound semiconductor etcher (ICP-RIE, Oxford Instruments System) with gas flow rates of $Cl_2$: Ar = 5 sccm: 30 sccm under 5 mTorr chamber pressure for 80 s. The double slits were obtained after the removal of ZEP 520A using remover PG. The resonators were finally defined by the third electron-beam writing process, followed by the same recipe for development,

and dry etched as before with a chamber pressure of 3 mTorr for 8 min. The final removal of ZEP 520A was performed using remover PG.

**Electrical connection to printed circuit boards (PCBs)**. We design two printed circuit boards (PCBs) to individually apply bias to each metasurface element across the antenna array. The sample is mounted on the first PCB, and 63 individual metasurface elements as well as the bottom contact are wire-bonded to 64 conducting pads on the PCB. Each conducting pad of the first PCB is then connected to an external pin on the second board. This PCB is capable of providing 64 independent voltages that can be individually controlled through the reference voltages derived by an external power supply (Keithley 2400). The second board has different configurations of voltage paths. By switching between different configurations, one can electrically change the grating periodicity of the metasurface.

## Data availability
The data that support the findings of this study are available from the authors on reasonable request; see author contributions for specific data sets.

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

## Acknowledgements

This work was supported by Samsung Electronics (P.C.W.), and NASA Early Stage Innovations (ESI) Grant 80NSSC19K0213 (H.A.A. & G.K.S.). The authors used facilities supported by the Kavli Nanoscience Institute (KNI). P.C.W. acknowledges the support from Ministry of Science and Technology, Taiwan (Grant numbers: 108-2112-M-006-021-MY3; 107-2923-M-001-010-MY3; 107-2923-M-006-004-MY3). P.C.W. also acknowledges the support in part by Higher Education Sprout Project, Ministry of Education to the Headquarters of University Advancement at National Cheng Kung University (NCKU). The authors deeply appreciate help in the form of the close reading of the manuscript and review responses by Rebecca Glaudell, Phil Jahelka, Kelly Mauser, Michael Kelzenberg, Joseph DuChene, and Haley Bauser. The authors also thank Artur Davoyan for useful discussions.

## Author contributions

P.C.W, R.A.P., and H.A.A. conceived the original idea. P.C.W. performed the numerical design, device fabrication, built up the optical setup, and performed the optical as well as high-speed measurements, analyzed numerical and experimental data, and wrote the paper; R.A.P. performed the numerical design, developed the dry etching process, and helped with the build-up of optical setup for measurement; G.K.S. helped with the sample fabrication, designed and build-up the PCB for individually electrical control of metasurface elements; W.-H.C. helped with the sample fabrication, data analysis, and optical measurement; R.S. developed the theoretical model for MQWs, performed calculations, and wrote the paper; M.G. helped with the high-speed measurement and data analysis; M.A. and D.L. helped with discussions; H.A.A. organized the project, designed experiments, analyzed the results, and prepare the papers. All authors discussed the results and commented on the paper.

## Additional information

**Competing interests:** The authors declare no competing interests.

