## [Peer Review File · Nature Communications]

Reviewers' Comments:

Reviewer #1:

Remarks to the Author:

The manuscript is devoted to realization of electrically tunable metasurface based on III-V compound semiconducting MQW structures as resonant metasurface elements. The metasurface consists of an array of two-dimensional hybrid Mie-guiding mode resonators which exhibit an actively tunable optical response under applied bias in the near-infrared wavelength range. The amplitude and phase of the light reflected from the metasurface is shown to be continuously tuned by applying DC electric field across the MQW metasurface elements, with a tunable optical response from the quantum-confined Stark effect. The topic of research presented is novel, combination of the well-established MQW technology with active subwavelength nanoantennas seems modern and prospective. Two main experimental findings can be pointed out. The first one is detection of large relative reflectance modulation of the metasurface (~270%) and a continuous phase shift from 0° to 70° at the resonant wavelength of 917 nm. The second one is demonstration of the dynamic diffraction grating by electrically connected metasurface elements in groups of three. In this case, the application of external bias leads to the appearance of diffraction orders which can be treated as a beam steering.

The manuscript can be published in Nature Communications after minor revisions reflecting following points.

1. The operation frequency of the device shown. The authors stated that "Since the amplitude modulation is achieved via the electro-optic effect rather than charge carrier injection, the intrinsic modulation frequency of our device can be GHz or higher". However, amplitude of the external bias is relatively small (several volts) and I do not see any experimental difficulty in application of AC electric field in order to obtain the temporal performance of the metasurface by lock-in technique. It would be great if authors perform these experiments.
2. Beam steering is achieved by switching the diffraction orders of dynamic grating. The efficiency of such effect is relatively low. Is it possible to increase the ratio between diffracted and mirror reflected beams?
3. Please check the Ref 3.

Reviewer #2:

Remarks to the Author:

The authors experimentally demonstrate electro-optical tuning of an all-dielectric metasurface based on the quantum-confined Stark effect in III-V multiple quantum wells.

Tunable metasurfaces are a hot topic, and the employed tuning mechanism is novel in this context and has good technical potential. Also, generally the work is well written and presented. However, there are some technical issues and unclear aspects which need to be resolved. Therefore, the following points should be addressed before the paper can be published in Nature Communications:

-The design of the metasurface is not very well motivated. Why is such a complicated resonator geometry chosen, which supports a Mie resonance coupled to a guided mode resonance? Would this approach also work with more common resonant dielectric metasurface geometries, having e.g. electric and magnetic dipole Mie resonances? Also, the partially etched slits are asymmetrically positioned with respect to the larger resonators. Does this asymmetry serve a particular purpose? Finally, it would be helpful to add in the Supplementary Material a multipole analysis of the higher-order Mie modes, since from the mode profiles they cannot be identified with typical Mie resonances.

-In the abstract and the main body of the manuscript, relative reflectance modulations are quoted. However, if using the low reflectance value as the reference which appears in the denominator, the

relative reflectance modulation can get very high at wavelengths of low reflectivity. The 270% achieved in this work sounds a lot, but compared to the ideal modulator, which would have a value of infinity for this measure, it is a moderate tuning performance. Also, in the wavelength range where the relative reflectance modulation is highest, the maximum reflectance of the metasurface is below 50%, which is not sufficient for high-efficiency devices. It would be more useful to quote absolute reflectance values or the absolute reflectance modulation, also to allow fair comparison with other tunable metasurfaces.

-On page seven it is stated „Higher-order diffracted beams are absent since the period of the sample is subwavelength $p=910$ nm at 0 V bias.“ On the same page, it also says: „The dynamic diffraction pattern measurements have been performed at a wavelength of 917 nm...“. Thus, the wavelength is very close to the lattice constant. For perfectly normal incidence and reflectance, indeed no diffraction should occur (at least in reflection, which is considered here). However, for slightly tilted incidence, as it effectively occurs when illuminating the sample with an objective or a lens, diffractive orders will occur starting from some critical angle. The authors should provide details of the measurement setup (in particular the NA of the focussing lens, if applicable) used to measure the data presented in Fig. 3b and discuss the occurrence of diffraction under the relevant conditions.

-The part on beam steering requires a more critical description and discussion. Usually, for a beam steering device or also the other applications mentioned in the outlook (metalenses with reconfigurable focal length, flat spatial light modulators etc), one would expect the strongest (typically the fundamental) reflected or transmitted order to be manipulated. In this work, it is only the first diffraction order, which carries only a small fraction of the reflected intensity, that can be manipulated. Also, there are first-order beams. Therefore, I would rather consider this structure a reconfigurable diffraction grating, not so much a beam steering device. For beam steering, I would also expect to see a (quasi) continuous variation of the angle, whereas only discrete angles can be achieved with the demonstrated device. In the light of these arguments, the authors may want to consider renaming the device, which also affects the title of this work.

Reviewer #3:

Remarks to the Author:

In this manuscript, the authors take advantage of the strong quantum-confined Stark effect in multiple quantum well (MQW) structures to control the effective refractive index of the micro-resonators and constructed metasurfaces based on those micro-resonators. The authors demonstrated the effective tuning of the refractive index, which resulted in substantial changes in reflectance and large shifts of the phase of the reflected beam. The authors also showed dynamic beam steering by controlling the refractive index distribution in the micro-resonators to achieve the modulation of the effective periodicity of the grating structure. But there are several issues that need to be resolved. I would recommend publication of this manuscript if the following questions are successfully addressed.

1. The authors claimed that the resonance studied in this work is from the coupling of a Mie resonance and a guided mode resonance. But in most cases when two resonances couple to each other, there will be interferences and mode-splitting or Fano-type of resonances will be observed. The mode studied here does not show those behaviors, why?
2. The authors concluded that the modulation is from pure electro-optic effect. How did they exclude the contribution of the carrier injection? Some detailed analysis will be very helpful to clarify this issue.
3. The authors showed a -1° average phase shift at a wavelength of 924nm. How much is the error level in the phase shift measurement? Although the -45% change in the reflection is not as

high as the 270% change at 917nm, it's still substantial. On the other hand, the -1° phase shift is quite small, which does not match the level of reflection change if we consider the K-K relation. Clarification is needed here. I noticed that there are some oscillations in the phase shift for 924nm wavelength. Is that just noise or real oscillation? Did the author go beyond the 0V to -10V range to find out the trend?

4. In the beam steering results, multiple diffraction beams were observed, especially in the second graph (red curve), there are two more beams with similar intensities as the claimed 1st order beams. What are they and why are they not considered as the 1st order beams?

5. In the sentence "where the first diffraction order angle becomes smaller as the metasurface periodicity of is reduced via electrical control" (page 8), there are either some words missing or the word "of" should be removed.

In this response letter, we offer a detailed point-by-point reply to each of the reviewers' comments. Our responses are shown *in blue*.

REVIEWERS' COMMENTS:

Reviewer: 1

The manuscript is devoted to realization of electrically tunable metasurface based on III-V compound semiconducting MQW structures as resonant metasurface elements. The metasurface consists of an array of two-dimensional hybrid Mie-guiding mode resonators which exhibit an actively tunable optical response under applied bias in the near-infrared wavelength range. The amplitude and phase of the light reflected from the metasurface is shown to be continuously tuned by applying DC electric field across the MQW metasurface elements, with a tunable optical response from the quantum-confined Stark effect. The topic of research presented is novel, combination of the well-established MQW technology with active subwavelength nanoantennas seems modern and prospective. Two main experimental findings can be pointed out. The first one is detection of large relative reflectance modulation of the metasurface (~270%) and a continuous phase shift from 0° to 70° at the resonant wavelength of 917 nm. The second one is demonstration of the dynamic diffraction grating by electrically connected metasurface elements in groups of three. In this case, the application of external bias leads to the appearance of diffraction orders which can be treated as a beam steering.

The manuscript can be published in *Nature Communications* after minor revisions reflecting following points.

We thank the reviewer for the positive evaluation of our work and the recommendation for publication after minor revisions.

1. The operation frequency of the device shown. The authors stated that “Since the amplitude modulation is achieved via the electro-optic effect rather than charge carrier injection, the intrinsic modulation frequency of our device can be GHz or higher”. However, amplitude of the external bias is relatively small (several volts) and I do not see any experimental difficulty in application of AC electric field in order to obtain the temporal performance of the metasurface by lock-in technique. It would be great if authors perform these experiments.

We thank the reviewer for this comment. We agree with the reviewer that high speed measurements can allow us to feasibly examine the modulation speed of MQW metasurfaces, which is an important parameter for tunable metadevices. To obtain the temporal performance and experimentally evaluate the modulation speed of our MQW metasurface, we performed

*amplitude modulation measurements with application of an AC electric field. The measured temporal response reveals that (see Supplementary Fig. 12b) the modulation speed of our fabricated MQW metasurface is on the order of MHz. We also theoretically estimate the RC delay in our MQW system (see Supplementary Note 7). The calculations show that the highest modulation speed can be on the order of MHz, which agrees well with the measured results. We emphasize that our device, which exploits the quantum confined Stark effect, can access its inherent limiting bandwidth operating with GHz modulation speed if the intrinsic RC delay is further minimized (Lewen, R., Irmscher, S., Westergren, U., Thylen, L. & Eriksson, U. Segmented transmission-line electroabsorption modulators. *J. Lightwave Technol.* 22, 172-179 (2004)). Nonetheless, MHz modulation is adequate for many applications such as, e.g., LiDAR. Based on these analyses, we added the following discussion and figures to the main text as well as the Supplementary Information.*

We revised the sentence in the main manuscript:

On Page 6

“... the intrinsic modulation frequency of our device can be MHz (see Supplementary Fig. 12) or higher^{31, 45, 46}”

We also added the following paragraph and figure to the Note 7 of Supplementary Material:

“We first theoretically estimate the modulation speed of the MQW metasurface. The conductivity of the p-GaAs (with doping level of 10^{19} cm^{-3}):

$$\sigma = qp\mu_p = 1.6 \times 10^{-19} \cdot 1 \times 10^{19} \cdot 67 \cong 107 \left[\frac{1}{\Omega \cdot \text{cm}} \right],$$

where q is the electron charge, p is the carrier concentration, $\mu_p \cong 67 \left[\frac{\text{cm}^2}{\text{V} \cdot \text{s}} \right]$ is the hole mobility⁵. The RC delay of the MQW system is

$$RC = \frac{L}{\sigma d_1 w_{\text{MQW}}} \frac{\varepsilon_0 \varepsilon_r L w_{\text{MQW}}}{d} = L^2 \frac{\varepsilon_0 \varepsilon_r}{\sigma d_1 d} = 1.816 \times 10^{-9} \text{ [s]},$$

where $L = 100 \text{ } \mu\text{m}$ and w_{MQW} is the length and width of the hybrid Mie-GM resonant structure respectively, d_1 is the thickness of p-GaAs, d is the thickness of MQW, ε_0 is the permittivity in vacuum, and $\varepsilon_r = 13.5$ is the dielectric constant^{6,7} of the MQW. Therefore, the estimated highest modulation frequency f is

$$f = \frac{1}{2\pi RC} = 87.66 \text{ [MHz]}.$$

To experimentally evaluate the modulation speed of our MQW metasurface, an AC electrical bias with frequencies of 0.1 MHz and 1 MHz is applied to the sample and a high-speed InGaAs detector is used to detect the temporal amplitude response (see Supplementary Fig. 12a). As

shown in Supplementary Fig. 12b, high-speed amplitude modulation with 0.1 MHz frequency bandwidth is performed. Modulation speed as high as 1 MHz is also demonstrated, which is on the order of the theoretical expectation. The difference between the theoretical calculation and the measured response is likely due to the deviation between parameters (conductivity, dielectric constant, *etc*) used in calculations and those of the real sample as well as the fact that the contact resistance is not taken into account in the theoretical estimation. In principle, the quantum confined Stark effect can yield the devices operating with GHz modulation speed if the intrinsic RC delay is further minimized, which can be accomplished by reducing the length of resonators, increasing the thickness of p-GaAs, *etc*. Note that the signal is distorted in the case of 1 MHz modulation speed due to the bandwidth limitation of the power amplifier.”

Supplementary Figure 12. Experimental performance of modulation speed measurement in all-dielectric MQW metasurface. (a) Schematic of the optical setup combined with a high-speed detector (Thorlabs DET10C) and an oscilloscope (Tektronix TDS 2001C). The AC electric field is applied from a function waveform generator (Keysight 33220A) combined with a power amplifier (Thorlabs HVA200). (b) The measured results of temporal response of hybrid Mie-GM metasurfaces. Blue curve: 1 MHz, red curve: 0.1 MHz. The wavelength of incident light is fixed at 917 nm.

with three additional references

5. Casey H. C., Stern F. Concentration-dependent absorption and spontaneous emission of heavily doped GaAs. *J. Appl. Phys.* **47**, 631-643 (1976).
6. Brennan K., Hess K. High field transport in GaAs, InP and InAs. *Solid State Electron.* **27**, 347-357 (1984).
7. Won-Pyo H., Bhattacharya P. K. High-field transport in InGaAs/InAlAs modulation-doped heterostructures. *IEEE T. Electron. Dev.* **34**, 1491-1495 (1987).

2. Beam steering is achieved by switching the diffraction orders of dynamic grating. The efficiency of such effect is relatively low. Is it possible to increase the ratio between diffracted and mirror reflected beams?

We thank the reviewer for bringing up this point. In principle, it is possible to increase the ratio between diffracted and mirror reflected beams by using the all-dielectric MQW metasurface design discussed in our work. To improve the directivity (which is defined as the peak intensity ratio between diffracted and mirror reflected beams), we need to have an almost constant reflectance accompanied with large phase shift when applying electrical bias. These conditions can be satisfied as long as the MQW system can provide a substantial change in the real part of the refractive index, Δn , to sufficiently shift the resonances, and maintain a small change in imaginary part of the refractive index (absorption), Δk , to enable a sharp resonance within operation wavelength range. As a result, the larger the figure of merit, $\Delta n/\Delta k$ of MQW, the larger tunable phase shift can be observed. The large tunable phase shift would yield the reflected beam with higher directivity. The $\Delta n/\Delta k$ of the MQW utilized in this work is 1-5 (see Google Patents, assignee US20150286078A1 2015). As a proof of concept, we designed a tunable metasurface with an asymmetric coupled quantum well (ACQW) which can possess a larger $\Delta n/\Delta k$ equal to 10-18 (see IEEE J. Quantum Elect. 34, 1197-1208 (1998)). The metasurface unit element is still based on the double-slit structure, as shown in Fig. R1a. After structural optimization, we found that about 200° phase shift with a Δn of 0.02 can be obtained (see Fig. R1b). The simulated far-field radiation patterns show that in this case, the intensity of steered beam is much higher than the intensity of the specularly reflected beam.

Figure R1. Simulated QW resonant metasurface with higher $\Delta n/\Delta k$. (a) A schematic for all-dielectric asymmetric coupling quantum well (ACQW) metasurface. The unit element dimensions are defined as follows: $w_c = 90$ nm, $w = 60$ nm, $g = 100$ nm, $t = 1230$ nm, and $h = 80$ nm. The periodicity p is 560 nm. (b) Simulated phase shift as a function of Δn of an ACQW resonant metasurface under an x-polarized normal illumination. (c) Simulated intensity of the scattered light in the far-field as a function of diffraction angle. The plotted diffracted light intensity is normalized to the light intensity at 0° . Right panel shows the corresponding phase profile for each case. The first-order diffracted

beam shows much higher intensity as compared to intensity of the specularly reflected beam. The phase shift and light intensity are plotted for a wavelength of 808.8 nm. Black arrows indicate the position of the first-order diffracted beams. Due to the spatial symmetry, only half of the radiation pattern is presented. The total number of unit elements is set at 120.

To address this issue, we revised the discussion of the last part of main text:

On Page 9 of the main text

“The performance of the proposed all-dielectric metasurface with hybrid Mie-GM resonance can be further improved by utilizing alternative QW systems which exhibit larger modulation of the real part of the refractive index and lower optical loss as compared to the QW used in the present work^{51, 52} (see Supplementary Note 13)”

with an additional reference

52. Hao F., Pang J. P., Sugiyama M., Tada K., Nakano Y. Field-induced optical effect in a five-step asymmetric coupled quantum well with modified potential. *IEEE J. Quantum Elect.* **34**, 1197-1208 (1998).

We also added a paragraph and a figure to the Supplementary Material.

Supplementary Note 13

Improvement of optical performance

“To improve the optical performance, the quantum well system has to provide a substantial change in the real part of the refractive index, Δn , to sufficiently shift the resonances, and maintain a small change in the imaginary part of refractive index (absorption), Δk , to enable a sharp resonance within the operating wavelength range. As a result, the larger the figure of merit, $\Delta n/\Delta k$ of a quantum well, the better optical performance can be achieved in the tunable metasurface. As a proof of concept, we designed a tunable metasurface with an asymmetric coupled quantum well (ACQW) which can possess a larger $\Delta n/\Delta k$ (about 10-18)¹¹. As a comparison, the $\Delta n/\Delta k$ of the utilized MQW in this work is 1-5 (see Ref. 1). The unit element is also based on the double-slit structure, as shown in Supplementary Fig. 18a. After structural optimization, we found that about 200° phase shift with a Δn of 0.02 can be obtained at a wavelength of 808.8 nm (see Supplementary Fig. 19b). We also numerically study the beam steering functionality using such an ACQW metasurface, which is realized by varying the periodicity of the metasurface (see Supplementary Fig. 19c). The simulated far-field radiation patterns even show that the intensity of the steered beam is much higher than the intensity of the specularly reflected beam when the utilized QW possesses larger $\Delta n/\Delta k$. These results indeed verify that the optical performance (in particular, directivity, which is defined as the peak intensity ratio between diffracted and mirror reflected beams) of tunable quantum well-based metasurfaces can be significantly improved when the quantum well system exhibits larger $\Delta n/\Delta k$. Since this is a proof-of-concept demonstration, the working wavelength here ($\lambda = 808.8$ nm) is slightly different from the one used in the main manuscript. By appropriately choosing a quantum well, we can shift the operation wavelength to the range of interest^{12, 13}.”

Supplementary Figure 19. Simulated QW resonant metasurface with higher $\Delta n/\Delta k$. (a) A schematic for all-dielectric asymmetric coupling quantum well (ACQW) metasurface. The unit element dimensions are defined as follows: $w_c = 90$ nm, $w = 60$ nm, $g = 100$ nm, $t = 1230$ nm, and $h = 80$ nm. The periodicity p is 560 nm. (b) Simulated phase shift as a function of Δn of an ACQW resonant metasurface under an x -polarized normal illumination. (c) Simulated intensity of the scattered light in the far-field as a function of diffraction angle. The plotted diffracted light intensity is normalized to the light intensity at 0° . Right panel shows the corresponding phase profile for each case. The first-order diffracted beam shows much higher intensity as compared to intensity of the specularly reflected beam. The phase shift and light intensity are plotted for a wavelength of 808.8 nm. Black arrows indicate the position of the first-order diffracted beams. Due to the spatial symmetry, only half of the radiation pattern is presented. The total number of unit elements is set at 120.

with three additional references

11. Hao F., Pang J. P., Sugiyama M., Tada K., Nakano Y. Field-induced optical effect in a five-step asymmetric coupled quantum well with modified potential. *IEEE J. Quantum Elect.* **34**, 1197-1208 (1998).
12. Xu Z., Wang C., Qi W., Yuan Z. Electro-optical effects in strain-compensated InGaAs/InAlAs coupled quantum wells with modified potential. *Opt. Lett.* **35**, 736-738 (2010).
13. Mohseni H., An H., Shellenbarger Z. A., Kwakernaak M. H., Abeles J. H. Enhanced electro-optic effect in GaInAsP–InP three-step quantum wells. *Appl. Phys. Lett.* **84**, 1823-1825 (2004)

3. Please check the Ref 3.

We thank the referee for pointing this out. The authors' names in Ref. 3 are checked and revised.

3. Haffner C., Heni C., Fedoryshyn Y., Niegemann J., Melikyan A., Elder D. L., et al. All-plasmonic Mach–Zehnder modulator enabling optical high-speed communication at the microscale. *Nat. Photon.* **9**, 525-528 (2015).

Reviewer: 2

The authors experimentally demonstrate electro-optical tuning of an all-dielectric metasurface based on the quantum-confined Stark effect in III-V multiple quantum wells.

Tunable metasurfaces are a hot topic, and the employed tuning mechanism is novel in this context and has good technical potential. Also, generally the work is well written and presented. However, there are some technical issues and unclear aspects which need to be resolved. Therefore, the following points should be addressed before the paper can be published in Nature Communications:

We are thankful to the reviewer for the positive evaluation of our work. We address the points brought up by the reviewer below.

1. The design of the metasurface is not very well motivated. Why is such a complicated resonator geometry chosen, which supports a Mie resonance coupled to a guided mode resonance? Would this approach also work with more common resonant dielectric metasurface geometries, having e.g. electric and magnetic dipole Mie resonances? Also, the partially etched slits are asymmetrically positioned with respect to the larger resonators. Does this asymmetry serve a particular purpose? Finally, it would be helpful to add in the Supplementary Material a multipole analysis of the higher-order Mie modes, since from the mode profiles they cannot be identified with typical Mie resonances.

We thank the reviewer for bringing up this point. First, we would like to mention that because of the relatively low index change of the experimentally implemented MQW (modulation of the real part of the refractive index is on the order of $\Delta n = 0.01$), fundamental electric and magnetic dipolar Mie resonances would not exhibit significant optical modulation under applied bias. This is because of the modest quality factor of the fundamental electric and magnetic dipolar Mie resonant modes. Therefore, in our work we utilize a resonator geometry supporting higher order modes, which exhibit a higher quality factor at resonance.

To motivate the choice of the utilized hybrid Mie-GM resonant metasurface element, we added the following sentence to the revised manuscript:

On Page 5

“Since our MQWs exhibit relatively modest refractive index change under applied bias, the designed metasurface element has to support high quality factor resonant mode near the semiconductor band edge in order to exhibit significant optical modulation under applied bias. The fundamental electric or magnetic dipole modes of typically utilized dielectric resonators do not possess sufficiently high quality factors.”

We also thank the reviewer for asking about the reasons behind the asymmetric placement of the partially etched slits. This asymmetry is from the misalignment of the two-step electron beam lithography processes. To study how asymmetric slit positioning affects the optical performance of the device, we calculate the reflectance of the metasurface for different positions of the topmost slits with respect to the center of the MQW slab, as shown in Fig. R2. Our simulations

show that the hybrid Mie-GM resonance exists in all three cases, $s = 0 \text{ nm}$, 40 nm , and 80 nm . However, this resonance blue-shifts when the spatial offset (s) is increased. To observe significant optical modulation, we need to ensure such high-quality resonance is spectrally located in the wavelength region where the utilized MQW exhibits the largest index modulation (that is, from 915 nm to 920 nm). Figure R2(b) indicates that the largest acceptable spatial offset s is about 80 nm (it is about 50 nm in our first fabricated MQW metasurface).

Figure R2. (a) Schematic for the hybrid Mie-GM resonant metasurface. The unit element dimensions are defined as follows: $w_c = 180 \text{ nm}$, $g = 100 \text{ nm}$, $t = 1230 \text{ nm}$, $h = 40 \text{ nm}$, and $p = 910 \text{ nm}$. The parameter s is defined as the spatial offset of the partially etched slits from the center of the underlying MQW slab. (b) Simulated reflectance spectrum for different spatial offsets s .

To address this issue, we added the following discussions and figure to the revised Supplementary Information

Supplementary Note 2

Asymmetry issue in hybrid Mie-GM resonant mode

“To study how asymmetric slit positioning affects the optical performance of the device (which is originally caused by the misalignment of the electron beam lithography process), we calculated the reflectance for different values of the offset parameter s (see Supplementary Fig. 5a). The offset parameter s is defined as the spatial displacement of center of the topmost partially etched slits with respect to the underlying MQW slab. Since the partially etched slits act as a light coupler and assist in excitation of guide-mode resonance, the generation of hybrid Mie-GM resonance is not significantly influenced by the structural asymmetry. Our simulations show that the hybrid Mie-GM resonance exists in all three cases, $s = 0 \text{ nm}$, 40 nm , and 80 nm . However, this resonance blue-shifts when the spatial offset (s) is increased. To observe significant optical modulation, we need to ensure such high-quality resonance is spectrally located in the wavelength region where the utilized MQW exhibits the largest index modulation (that is, from 915 nm to 920 nm). Supplementary Figure 5(b) indicates that the largest acceptable spatial offset s is about 80 nm (it is about 50 nm in our first fabricated MQW metasurface).”

Supplementary Figure 5. (a) Schematic for the hybrid Mie-GM resonant metasurface. The unit element dimensions are defined as follows: $w_c = 180$ nm, $g = 100$ nm, $t = 1230$ nm, $h = 40$ nm, and $p = 910$ nm. The parameter s is defined as the spatial offset of the partially etched slits from the center of the underlying MQW slab. (b) Simulated reflectance spectrum for different spatial offsets s .

Finally, we agree with the reviewer that the multipole decomposition analysis will be instrumental for understanding the high-order Mie-GM resonance excited within the MQW metasurface. Thus, we performed a calculation based on the charge-current expansion framework (*Phys. Rev. B* **89**, 205112 (2014); *ACS Nano* **12**, 1920-1927 (2018); *Light: Science & Applications* **7**, 17158 (2018)). To clearly classify the contribution of each electromagnetic multipole, fundamental (electric dipole) and high-order modes (magnetic dipole, electric and magnetic quadrupoles and octupoles) are considered in the calculation. According to the calculated results, the magnetic octupolar mode dominates the spectral range of interest. This is consistent with the simulated field profiles shown in Figs. 2c and 2d.

To address this issue, we added the following discussion in revised manuscript.

On Page 5

“The calculated field profiles show that at a wavelength of 915.9 nm, the metasurface element supports a high-order Mie resonance (left images in Figs. 2c and 2d). The multipole decomposition analysis⁴⁸⁻⁵⁰ shows that the supported high-order Mie resonant mode is dominated by the magnetic octupolar mode (see Supplementary Note 2).”

with three additional references:

48. Savinov V., Fedotov V. A., Zheludev N. I. Toroidal dipolar excitation and macroscopic electromagnetic properties of metamaterials. *Phys. Rev. B* **89**, 205112 (2014).
49. Wu P. C., Liao C. Y., Savinov V., Chung T. L., Chen W. T., Huang Y.-W., et al. Optical anapole metamaterial. *ACS Nano* **12**, 1920-1927 (2018).
50. Zhu A. Y., Chen W. T., Zaidi A., Huang Y.-W., Khorasaninejad M., Sanjeev V., et al. Giant intrinsic chiro-optical activity in planar dielectric nanostructures. *Light: Science & Applications* **7**, 17158 (2018).

We also added the following discussions and figure to the revised Supplementary information.

Supplementary Note 2

Calculation of multipole decomposition

“To further analyze the modes supported by the MQW metasurface, we performed a multipole decomposition analysis using the charge-current expansion framework²⁻⁴. In our calculations, we account for contributions of electric and magnetic dipoles, quadrupoles, and octupoles.

Supplementary Figure 4 shows results of the radiated electric field intensity contributed from various electromagnetic multipoles. For simplicity, only the first three leading terms are shown. As expected from the field distribution presented in Figs. 2c and 2d, the magnetic octupole (which can be interrupted as a high-order Mie resonance) plays a significant role in the optical response under x -polarized illumination.”

Supplementary Figure 4. Simulated z -component of far-field electric intensity for electromagnetic multipoles. For simplicity, only the first three leading terms are presented.

with three additional references:

2. Savinov V., Fedotov V. A., Zheludev N. I. Toroidal dipolar excitation and macroscopic electromagnetic properties of metamaterials. *Phys. Rev. B* **89**, 205112 (2014).
3. Wu P. C., Liao C. Y., Savinov V., Chung T. L., Chen W. T., Huang Y.-W., et al. Optical anapole metamaterial. *ACS Nano* **12**, 1920-1927 (2018).
4. Zhu A. Y., Chen W. T., Zaidi A., Huang Y.-W., Khorasaninejad M., Sanjeev V., et al. Giant intrinsic chiro-optical activity in planar dielectric nanostructures. *Light: Science & Applications* **7**, 17158 (2018).

2. In the abstract and the main body of the manuscript, relative reflectance modulations are quoted. However, if using the low reflectance value as the reference which appears in the denominator, the relative reflectance modulation can get very high at wavelengths of low reflectivity. The 270% achieved in this work sound a lot, but compared to the ideal modulator, which would have a value of infinity for this measure, it is a moderate tuning performance. Also, in the wavelength range where the relative reflectance modulation is highest, the maximum reflectance of the metasurface is below 50%, which is not sufficient for high-efficiency devices. It would be more useful to quote absolute reflectance values or the absolute reflectance modulation, also to allow fair comparison with other tunable metasurfaces.

We thank the reviewer for bringing up this point. We agree with the reviewer’s comment that the relative optical intensity (reflectance or transmittance) modulation can be infinity (for the cases where the reference intensity is close to zero). We also agree that reporting absolute reflectance modulation can allow fair comparison with other tunable metasurfaces. Based on the reviewer’s suggestion, we provided the measured results of absolute reflectance modulation in

Supplementary Fig. 10. We observe about +20% and -30% absolute reflectance modulation [defined as $R(V_a \neq 0 \text{ V}) - R(V_a = 0 \text{ V})$] at wavelengths of 917 nm and 924 nm, respectively. Although they are quantitatively comparable, we would like to point out that both significant phase modulation and optical diffraction switching are experimentally observed only at a wavelength of 917 nm (see Figs. 3d, 4, and Supplementary Note 8), which also yields higher relative reflectance modulation. In a broader context of work on tunable metasurfaces, the relative change of optical intensity (reflectance or transmittance) has been widely used to quantitatively evaluate the optical modulation capability of tunable metasurfaces (for example, see *Nano Lett.* 14, 6526-6532 (2014); *Nat. Nanotechnol.* 8, 252-255 (2013); *ACS Nano* 9, 4308-4315 (2015), etc.), especially for those cases in which the base line reflectance $R(V_a = 0 \text{ V})$ is greater than 1% (which is our case here). As a result, we believe that simultaneous reporting of both relative and absolute reflectance modulation values enables comprehensive evaluation of the performance of tunable metasurfaces. We added the following discussion and results to the revised manuscript text and Supplementary information:

On Page 6 in the main text:

“It is worth mentioning that we observe about +20% and -30% absolute reflectance modulation [defined as $R(V_a \neq 0 \text{ V}) - R(V_a = 0 \text{ V})$] at wavelengths of $\sim 917 \text{ nm}$ and $\sim 924 \text{ nm}$, respectively (see Supplementary Figure 10). Although these values are quantitatively comparable, the large phase modulation and diffracted beam switching can only be observed at a wavelength of 917 nm, when high-quality resonance is present (which shows higher relative reflectance modulation), as can be seen in the following sections.”

In the Supplementary Material:

Supplementary Note 5

Measured absolute reflectance modulation

Supplementary Figure 10. Measured absolute reflectance modulation ΔR of the hybrid Mie-GM resonant metasurface.

- On page seven it is stated “Higher-order diffracted beams are absent since the period of the sample is subwavelength $p=910 \text{ nm}$ at 0 V bias.” On the same page, it also says: “The dynamic diffraction pattern measurements have been performed at a wavelength of 917 nm...”. Thus, the wavelength is very close to the lattice constant. For perfectly normal

incidence and reflectance, indeed no diffraction should occur (at least in reflection, which is considered here). However, for slightly tilted incidence, as it effectively occurs when illuminating the sample with an objective or a lens, diffractive orders will occur starting from some critical angle. The authors should provide details of the measurement setup (in particular the NA of the focussing lens, if applicable) used to measure the data presented in Fig. 3b and discuss the occurrence of diffraction under the relevant conditions.

We thank the reviewer for his/her insightful comment. We agree with the reviewer that diffraction can contribute to the spectral features when structures are excited under oblique illumination at angles larger than a critical angle. The details of the optical setup used for the measurement of far-field radiation patterns are provided in Fig. 4c and the corresponding figure caption. Based on the used numerical aperture of the objective, the largest incident angle is about 16° when the laser beam is tightly focused onto the MQW samples. To minimize the effect of diffraction caused by the non-zero angle of incidence, we intentionally slightly defocused the laser beam impinging on the MQW metasurface during the optical measurements of far-field radiation pattern.

To clarify this point, we performed numerical simulations of the far-field radiation patterns for the MQW metasurface at 0V with different angles of incidence. As shown in Supplementary Fig. 11a, we found that strong optical diffraction appears even when the incident angle is 5° . However, those diffracted beam intensities are high compared to the intensity of the zero-order beam, which is not observed in our measurements. This can be attributed to two different factors; first, their diffraction angles are too large to be collected (based on the numerical aperture of the objective we used, the largest angle collected is about 16°) and second, the incident angle is almost zero in the real case (because we did not tightly focus the laser beam in such measurements). As a result, we did not see such diffracted beams in our measurement at 0V.

To further verify the influence of non-zero incident angle caused diffraction on the far-field radiation pattern, we performed other simulations which numerically demonstrate the active switching of the first-order diffracted beam at different angles of incidence. As shown in Supplementary Fig. 11b, for both cases of normal and oblique illuminations, we can see that the first-order diffracted beams appear only when the electrical bias is non-zero. In addition, the intensity of the diffracted beams are very high when the incident angle is greater than 5° , which is in conflict with our measurement results. As a result, we conclude that the MQW metasurface is under almost normal illumination (θ_{in} is between 0° and 5°), which minimizes the influence of diffraction caused by non-normal illumination in the real case. Moreover, no higher-order diffracted beams appear within the angular range of interest in all simulated results.

We revised this sentence in the manuscript:

On Page 7

“Higher-order diffracted beams are absent since the period of our metasurface, $p = 900$ nm, is subwavelength at 0 V bias (see details in Supplementary Note 6).”

We also added a detailed discussion of the influence of oblique incidence to the revised Supplementary information as follows:

Supplementary Note 6

Influence of oblique illumination on the optical diffraction

“Because of the slight difference in structural period (910 nm) and laser wavelength (917 nm), optical diffraction can influence the far-field radiation pattern when the incident angle is non-zero. To clarify this point, we performed numerical simulations of the far-field radiation patterns for the MQW metasurface at 0V with different angles of incidence. As shown in Supplementary Fig. 11a, we found that strong optical diffraction appears even when the incident angle is 5°. However, those diffracted beams' intensities are high compared as to the intensity of the zero-order beam, which is not observed in our measurements. This can be attributed to two different reasons; first, their diffraction angles are too large to be collected (based on the numerical aperture of the objective we used, the largest angle collected is about 16°) and second, the incident angle is almost zero in the real case. To experimentally eliminate this effect, we intentionally slightly defocused the laser beam onto the MQW metasurface when performing the far-field radiation measurements to minimize the incident angle.

To further verify the influence of non-zero incident angle on the far-field radiation pattern, we performed other simulations which numerically demonstrate the active switching of the first-order diffracted beam at different angles of incidence. As shown in Supplementary Fig. 11b, the intensity of the first-order diffracted beams are much higher as compared with the specularly reflected beam when the incident angle is greater than 5°, which is in conflict with our measurement results shown in Figs. 4d and 4e. As a result, we conclude that the MQW metasurface is under almost normal illumination ($0^\circ \leq \theta_{in} \leq 5^\circ$), and the non-zero incident angle caused optical effect is fairly small in the real case. It is worth noting that the first-order diffracted beams can only be observed when electrical bias is applied, indicating that the demonstration of active switching of first-order diffracted beam is still valid even when the MQW metasurface is under oblique illumination.”

Supplementary Figure 11. (a) Simulation of far-field radiation patterns under oblique illumination without electrical bias. Strong diffraction can be observed when the incident angle is greater than 5°. (b) Simulation of active switching of first-order diffraction for the case of normal (left panel) and oblique (middle and right panels) illumination. Overall, the diffracted beams show much stronger intensity when incident angle is greater than 5°. The incident wavelength is fixed at 917 nm.

We also added details of the experimental setup used for spectral measurement and a couple of sentences to the main text and Supplementary as follows.

On Page 6 in the main text:

“Figure 3b shows the measured reflectance spectra of the fabricated metasurface under different applied biases (see Supplementary Fig. 9 for details of optical setup).”

Supplementary Note 5

Optical setup for measurement of reflectance spectrum

“To optically characterize the reflectance of the MQW metasurface, we utilized a coherent NIR laser beam (Toptica Photonics CTL 950) as a light source and a power meter as a detector. An uncollimated white light source from a halogen lamp is used to visualize the sample surface. When measuring the reflectance spectra, the laser beam was focused using a long working distance objective with 10× magnification and 0.28 numerical aperture.”

Supplementary Figure 9. Schematic of optical setup used for spectral measurement. M: mirror; ND: neutral density filter (Thorlabs NDC-50C-4M); I: iris; L: lens; P: linear polarizer (Thorlabs LPNIR100-MP); BS: beam splitter (Thorlabs CCM1-BS014); O: objective (Mitutoyo 10× magnification with 0.28 numerical aperture); PM: power meter.

4. The part on beam steering requires a more critical description and discussion. Usually, for a beam steering device or also the other applications mentioned in the outlook (metalenses with reconfigurable focal length, flat spatial light modulators *etc*), one would expect the strongest (typically the fundamental) reflected or transmitted order to be manipulated. In this work, it is only the first diffraction order, which carries only a small fraction of the reflected intensity, that can be manipulated. Also, there are first-order beams. Therefore, I would rather consider this structure a reconfigurable diffraction grating, not so much a beam steering device. For beam steering, I would also expect to see a (quasi) continuous variation of the angle, whereas only discrete angles can be achieved with the demonstrated device. In the light of these arguments, the authors may want to consider renaming the device, which also affects the title of this work.

We thank the reviewer for bringing up this issue. However, we respectfully disagree with the reviewer’s comment. We agree that in this work, the intensity of the steered beam is small. However, we would like to note that the definition of an optical component/device should be determined by its mode of functionality rather than operation efficiency. As noted in response to the second comment from the first reviewer, the optical performance of the demonstrated tunable metasurface is actually limited by a non-optimal choice of the quantum well material, rather than the metasurface approach. Based on the discussions and simulated results shown in Supplementary Note 13, we indeed show that the directivity (which is defined as the peak intensity ratio between diffracted and mirror reflected beams) of our double-slit metasurface can be significantly improved when the quantum well system possesses larger $\Delta n/\Delta k$.

The following discussions and simulated results are added to the Supplementary Material.

Supplementary Note 13

Improvement of optical performance

“To improve the optical performance, the quantum well system has to provide a substantial change in the real part of the refractive index, Δn , to sufficiently shift the resonances, and maintain a small change in the imaginary part of refractive index (absorption), Δk , to enable a sharp resonance within the operating wavelength range. As a result, the larger the figure of merit, $\Delta n/\Delta k$ of a quantum well, the better optical performance can be achieved in the tunable metasurface. As a proof of concept, we designed a tunable metasurface with an asymmetric coupled quantum well (ACQW) which can possess a larger $\Delta n/\Delta k$ (about 10-18)¹¹. As a comparison, the $\Delta n/\Delta k$ of the utilized MQW in this work is 1-5 (see Ref. 1). The unit element is also based on the double-slit structure, as shown in Supplementary Fig. 18a. After structural optimization, we found that about 200° phase shift with a Δn of 0.02 can be obtained at a wavelength of 808.8 nm (see Supplementary Fig. 19b). We also numerically study the beam steering functionality using such an ACQW metasurface, which is realized by varying the periodicity of the metasurface (see Supplementary Fig. 19c). The simulated far-field radiation patterns even show that the intensity of the steered beam is much higher than the intensity of the specularly reflected beam when the utilized QW possesses larger $\Delta n/\Delta k$. These results indeed verify that the optical performance (in particular, directivity, which is defined as the peak intensity ratio between diffracted and mirror reflected beams) of tunable quantum well-based metasurfaces can be significantly improved when the quantum well system exhibits larger $\Delta n/\Delta k$. Since this is a proof-of-concept demonstration, the working wavelength here ($\lambda = 808.8$ nm) is slightly different from the one used in the main manuscript. By appropriately choosing a quantum well, we can shift the operation wavelength to the range of interest^{12, 13}.”

Supplementary Figure 19. Simulated QW resonant metasurface with higher $\Delta n/\Delta k$. (a) A schematic for all-dielectric asymmetric coupling quantum well (ACQW) metasurface. The unit element dimensions are defined as

follows: $w_c = 90$ nm, $w = 60$ nm, $g = 100$ nm, $t = 1230$ nm, and $h = 80$ nm. The periodicity p is 560 nm. (b) Simulated phase shift as a function of Δn of an ACQW resonant metasurface under an x -polarized normal illumination. (c) Simulated intensity of the scattered light in the far-field as a function of diffraction angle. The plotted diffracted light intensity is normalized to the light intensity at 0° . Right panel shows the corresponding phase profile for each case. The first-order diffracted beam shows much higher intensity as compared to intensity of the specularly reflected beam. The phase shift and light intensity are plotted for a wavelength of 808.8 nm. Black arrows indicate the position of the first-order diffracted beams. Due to the spatial symmetry, only half of the radiation pattern is presented. The total number of unit elements is set at 120.

with three additional references

11. Hao F., Pang J. P., Sugiyama M., Tada K., Nakano Y. Field-induced optical effect in a five-step asymmetric coupled quantum well with modified potential. *IEEE J. Quantum Elect.* **34**, 1197-1208 (1998).
12. Xu Z., Wang C., Qi W., Yuan Z. Electro-optical effects in strain-compensated InGaAs/InAlAs coupled quantum wells with modified potential. *Opt. Lett.* **35**, 736-738 (2010).
13. Mohseni H., An H., Shellenbarger Z. A., Kwakernaak M. H., Abeles J. H. Enhanced electro-optic effect in GaInAsP-InP three-step quantum wells. *Appl. Phys. Lett.* **84**, 1823-1825 (2004)

In addition, we also want to point out that when utilizing grating approach, there are always sets of discrete angles, which are accessible (see Proc. IEEE 97, 1078-1096, 2009). In the above-mentioned literature, they use the term "beam steering" even if the steering angles they can access are discrete. The angle step will be much smaller when the range of the steering angles is decreased, which can make the variation of angle more continuous. This can be experimentally realized when more unit elements are involved in the metasurface fabrication and printed circuit board (PCB) design.

Thus, since our device performance is limited by the QW design and electrical control practicalities (which have been addressed in the literature) rather than any conceptual limitations, we believe that the second metasurface demonstrated in this work can still be categorized as a beam-steering device.

Reviewer: 3

In this manuscript, the authors take advantage of the strong quantum-confined Stark effect in multiple quantum well (MQW) structures to control the effective refractive index of the micro-resonators and constructed metasurfaces based on those micro-resonators. The authors demonstrated the effective tuning of the refractive index, which resulted in substantial changes in reflectance and large shifts of the phase of the reflected beam. The authors also showed dynamic beam steering by controlling the refractive index distribution in the micro-resonators to achieve the modulation of the effective periodicity of the grating structure. But there are several issues that need to be resolved. I would recommend publication of this manuscript if the following questions are successfully addressed.

We are thankful to the reviewer for the positive evaluation of our work. We address the points brought up by the reviewer below.

1. The authors claimed that the resonance studied in this work is from the coupling of a Mie resonance and a guided mode resonance. But in most cases when two resonances couple to each other, there will be interferences and mode-splitting or Fano-type of resonances will be observed. The mode studied here does not show those behaviors, why?

We thank the reviewer for bringing up this point. We agree that a mode-splitting profile should be observed when two resonances couple to each other. From the measured reflectance spectrum shown in Fig. 3b, we would expect to see the second resonant dip appearing at a wavelength shorter than 915 nm under zero bias. To verify that the mode-splitting occurs in our case, we performed a simulation in which the wavelength ranged from 900 nm to 950 nm, ensuring the observation of the second resonant dip. According to the simulation results (as shown in Fig. R3), we do observe the second resonant mode at a wavelength of 902.4 nm, which is consistent with the physical picture of mode-splitting.

Figure R3. Simulated reflectance spectrum of a Mie-GM resonant metasurface under an x-polarized normal illumination. The structural parameters are identical to the ones described in Fig. 2.

*We added the following discussion to the main text:
On Page 6*

“Note that the coupling of two resonant modes normally results in mode splitting. In our case, the mode splitting can be seen when extending the simulation range to the shorter wavelengths (see Supplementary Note 4).”

We also added the following figure to the Supplementary Material:

Supplementary Note 4

Mode splitting in the hybrid Mie-GM resonant metasurface

Supplementary Figure 8. Simulated reflectance spectrum of a Mie-GM resonant metasurface under an x -polarized normal illumination. Here, we extend the wavelength range to shorter wavelengths as compared to the wavelength range over which we performed our simulations. The structural parameters are identical to the ones described in Fig. 2.

2. The authors concluded that the modulation is from pure electro-optic effect. How did they exclude the contribution of the carrier injection? Some detailed analysis will be very helpful to clarify this issue.

*We thank the reviewer for the insightful comment. To access carrier-induced refractive index change in InGaAs and GaAs based III-V semiconductor compound, the injected current density has to be on the order of kA/cm^2 (see *Appl. Phys. Lett.* 50, 141-142 (1987), *IEEE J. Sel. Top. Quant.* 1, 408-415 (1995), and *Appl. Phys. Lett.* 45, 836-837 (1984)), which is much higher than the current density in our case (mA/cm^2). Consequently, we concluded that the optical modulation in our hybrid Mie-GM resonant metasurface is caused by an electro-optic effect. To clarify this issue, we added following description in the main text:*

On Page 6

“...since the amplitude modulation is achieved via the electro-optic effect rather than charge carrier injection (due to the low leakage current density in our samples, see Supplementary Note 11)...”

We also added following discussions to the Supplementary Note 11:

“Supplementary Fig. 17 shows the measured leakage current density of two MQW resonators. To avoid the dielectric breakdown, we applied a moderate bias ranging from 0 V to -10 V. For both of our samples, the measured current density is on the order of mA/cm^2 , which is much

lower than the current density values (on the order of of $\sim \text{kA}/\text{cm}^2$) necessary for observation for carrier-induced refractive index change in GaAs-based III-V semiconductor compounds⁸⁻¹⁰.”

with three additional references:

8. Dutta N. K., Olsson N. A., Tsang W. T. Carrier induced refractive index change in AlGaAs quantum well lasers. *Appl. Phys. Lett.* **45**, 836-837 (1984).
9. Ozaki S., Adachi S. Spectroscopic ellipsometry and thermorefectance of GaAs. *J. Appl. Phys.* **78**, 3380-3386 (1995).
10. Jong-In S., Yamaguchi M., Delansay P., Kitamura M. Refractive index and loss changes produced by current injection in InGaAs(P)-InGaAsP multiple quantum-well (MQW) waveguides. *IEEE J. Sel. Top. Quant.* **1**, 408-415 (1995).

3. The authors showed a -1° average phase shift at a wavelength of 924nm. How much is the error level in the phase shift measurement? Although the -45% change in the reflection is not as high as the 270% change at 917nm, it's still substantial. On the other hand, the -1° phase shift is quite small, which does not match the level of reflection change if we consider the K-K relation. Clarification is needed here. I noticed that there are some oscillations in the phase shift for 924nm wavelength. Is that just noise or real oscillation? Did the author go beyond the 0V to -10V range to find out the trend?

We thank the reviewer for bringing up these points. The oscillations in the measured phase shift are caused by the noise in the captured CCD images. To obtain a more accurate evaluation of the measured phase shift, we measure the phase shifts by illuminating different parts of our MQW metasurface. In revised Fig. 3d and Supplementary Fig. 15c, each point represents the average phase shift measured from four different positions on the sample while the error bars indicate the standard deviation of the obtained four data points. The results and corresponding captions are revised in the main manuscript and Supplementary Material.

Main manuscript, Figure 3

Figure 3 | Experimental verification of optical modulation in MQW resonators. (a) SEM image of multiple quantum well-based Mie resonators with double slits. (b) Measured reflectance spectra of the Mie resonator array for different applied bias voltages. The incoming light is polarized perpendicularly to the multiple quantum well stripes. (c) Measured relative reflectance of the hybrid Mie-GM resonant metasurface as a function of wavelength and applied voltage. We consider the wavelength range from 915 nm to 925 nm with a step of 1 nm. (d) Measured phase modulation at two different wavelengths. Red: 917 nm, blue: 924 nm. Each data point corresponds to an average phase shift measured at four different positions on the sample while each error bar indicates the standard deviation of the obtained four data points.

Supplementary Fig. 15

Supplementary Figure 15. (a) Measured reflectance spectra of the second Mie resonator array for different applied bias voltages. The incoming light is polarized perpendicularly to the MQW stripes. (b) Measured relative reflectance of the second hybrid Mie-GM resonant metasurface as a function of wavelength and applied voltage. We consider the wavelength range from 915 nm to 925 nm with a step of 1 nm. Here, the reflectance of resonator under a -10 V bias is utilized as the reference. (c) Measured phase modulation at two different wavelengths. Red: 917 nm, blue: 924 nm. Each data point corresponds to an average phase shift measured at four different positions on the sample while each error bar indicates the standard deviation of the obtained four data points.

In addition, we would like to clarify that -1° is the average of 11 data points from different applied biases. For our first sample, at a wavelength of 924 nm and at an applied bias of -10 V we observe a relative reflectance modulation of -45% accompanied by a phase shift is 12° which is non-negligible (see Fig. 3). Thus, the measured results do not violate the K-K relation. To avoid confusion, we modified the discussion in the main text as follows:

On Page 7

“For example, at a wavelength of 924 nm, the largest phase shift reduces to 12° , which we observed at an applied bias of -10 V. This modest phase shift is also accompanied by a weaker relative reflectance modulation of -45% (see Fig. 3c).”

Finally, as discussed in Supplementary Note 11, to avoid the dielectric breakdown, the applied electrical bias ranges only from 0 V to -10 V. The measured phase shift is a much smoother function of applied bias (see Fig. 3d and Supplementary Fig. 15c) when we account for the data acquired from the multiple positions on the sample while processing the measurement results.

4. In the beam steering results, multiple diffraction beams were observed, especially in the second graph (red curve), there are two more beams with similar intensities as the claimed 1st order beams. What are they and why are they not considered as the 1st order beams?

We thank the referee for pointing this out. The sidelobes that appear around the zero-order diffraction are from the finite aperture effect, which happens when the laser spot size is larger than the size of the beam steering metasurface. This can also be observed in the simulation results shown in Supplementary Figure 16a. To clarify the position of the first diffraction order, we provided the simulations with larger number of unit elements, as shown in Supplementary Figure 16b. By comparing Fig. 5 with Supplementary Fig. 16, we can conclude which peaks are due to the first-order diffraction (the ones indicated by black arrows).

*We added the following discussions into the main manuscript to address the reviewer's concern.
On Page 8*

“Note that the sidelobes appear around the zeroth-order diffraction beam are from the finite aperture effect, which can be seen in both measured and simulated results. By characterizing the measured and simulated far-field radiation patterns with a larger total number of unit elements (Fig. 5e and Supplementary Fig. 16), the first-order diffraction peaks can be picked out, as indicated by black arrows in Fig. 5e.”

We also added the following sentence into the caption of Supplementary Figure 16 to clarify the illumination condition used for beam steering simulations:

Supplementary Figure 16 caption

“...In order to take the finite aperture effect into account, the top hat is utilized as the illumination condition when processing the far field projection in Lumerical FDTD simulation.”

5. In the sentence "where the first diffraction order angle becomes smaller as the metasurface periodicity of is reduced via electrical control" (page 8), there are either some words missing or the word "of" should be removed.

We thank the referee for pointing this out. We revised this sentence in the main text as following:

“...where the first-order diffraction angle becomes smaller as the metasurface periodicity is increased via electrical control”

Reviewers' Comments:

Reviewer #1:

Remarks to the Author:

The authors properly revised the manuscript accounting referee's critics. The manuscript can be published in the revised form.

Reviewer #2:

Remarks to the Author:

The authors have addressed all my critical points and I can now recommend the manuscript for publication.

Reviewer #3:

Remarks to the Author:

The authors have successfully addressed the questions I raised in the previous review. Therefore I recommend it to be published in Nature Communications in its current form.